# Maternal SARS-CoV-2 Infection at Delivery Increases IL-6 Concentration in Umbilical Cord Blood

**DOI:** 10.3390/jcm12175672

**Published:** 2023-08-31

**Authors:** Katarzyna Kosińska-Kaczyńska, Beata Rebizant, Hanna Czeszko-Paprocka, Agata Bojdo, Maciej Przybylski, Katarzyna Chaberek, Agnieszka Lewandowska, Iwona Szymusik, Robert Brawura-Biskupski-Samaha

**Affiliations:** 1Department of Obstetrics, Perinatology and Neonatology, Center of Postgraduate Medical Education, 01-809 Warsaw, Poland; katarzyna.kosinska-kaczynska@cmkp.edu.pl (K.K.-K.); agata.bojdo@bielanski.med.pl (A.B.); chaberek.katarzyna@gmail.com (K.C.); agnieszka.lewandowska@cmkp.edu.pl (A.L.); iwona.szymusik@cmkp.edu.pl (I.S.); robertsamaha@gmail.com (R.B.-B.-S.); 2Central Analytical Laboratory, Warsaw Infectious Diseases Hospital, 01-201 Warsaw, Poland; hpaprocka@zakazny.pl; 3Department of Medical Microbiology, Medical University of Warsaw, 02-091 Warsaw, Poland; maciej.przybylski@wum.edu.pl

**Keywords:** COVID-19, SARS-CoV-2, pregnancy, interleukin 6, amniotic fluid, umbilical cord blood

## Abstract

Background: SARS-CoV-2 infection in pregnant women may induce inflammation within the amniotic cavity and/or an increase in proinflammatory cytokines in fetal circulation. The aim was to investigate levels of IL-6 in maternal blood, umbilical cord blood, and amniotic fluid in pregnant women infected with SARS-CoV-2 at delivery. Methods: A single-center prospective observational case–control study of pregnant women diagnosed with SARS-CoV-2 infection at delivery was conducted. A total of 48 infected and 42 healthy women had IL-6 concentrations measured in their blood, amniotic fluid, and umbilical cord blood. Results: The concentrations of IL-6 in maternal blood and amniotic fluid were similar in the study and control groups, while umbilical cord blood concentrations were significantly higher in SARS-CoV-2-positive women. The umbilical cord blood IL-6 concentration was related to composite neonatal morbidity. Conclusions: Maternal SARS-CoV-2 infection in pregnant women at delivery increases umbilical cord blood IL-6 concentration. The correlation between maternal and umbilical blood concentrations indicates a possibility of passage of IL-6 through the placenta. Perinatal alterations resulting from maternal SARS-CoV-2 infection at delivery carry a risk of impacting the health of infants even in asymptomatic course of infection.

## 1. Introduction

Severe acute respiratory syndrome coronavirus-2 (SARS-CoV-2) was identified on 7 January 2020, and after two months the World Health Organization declared the novel coronavirus outbreak a global pandemic [1]. At that time, coronavirus disease 2019 (COVID-19) was one of the leading causes of maternal death worldwide, causing pneumonia, acute respiratory distress syndrome, and multiple organ failure. Although COVID-19 is usually asymptomatic or mild-symptomatic, pregnant women are at increased risk of respiratory failure or death [2,3]. SARS-CoV-2 infection during pregnancy is also related to an increased risk of miscarriage, preeclampsia, preterm delivery, and intrauterine fetal demise [4,5].

Interleukin 6 (IL-6) is a multifunctional cytokine. It regulates immune responses, inflammation, hematopoesis, and also trophoblast proliferation, invasion, and differentiation [6]. It is produced by macrophages, monocytes, fibroblasts, endothelial cells, and trophoblasts [7]. This cytokine is a major mediator of organism response to infection or tissue injury. It plays a role in T-cell activation, induces differentiation of cytotoxic T cells, and induces antibody secretion by B cells [8]. In severe cases of COVID-19, a cytokine storm has been observed [9]. The virus enters the cell and provokes an immune response with massive inflammatory cytokine production. SARS-CoV-2 can rapidly activate pathogenic Th1 cells to secrete proinflammatory cytokines, such as granulocyte-macrophage colony-stimulating factor (GM-CSF) and IL-6 [10]. The cytokine storm is characterized by a high expression of IL-6 and TNF-α [10]. High levels of serum IL-6 were observed in patients with severe and critical COVID-19 in comparison to asymptomatic patients, and the highest values were observed in non-survivors [11].

The virus was detected using polymerase chain reaction (PCR) in the placenta, umbilical cord blood, or neonatal blood collected within the first 12 h after birth, or amniotic fluid collected prior to the rupture of membranes, which supports the potential of congenital, intrapartum, and postnatal maternal–fetal–neonatal SARS-CoV-2 infections [12,13]. In infected newborns, mild to moderate symptoms associated with SARS-CoV-2 including cough, respiratory distress, fever, and pneumonia may be present [14].

According to Romero, preterm delivery is a symptom with many causes, among others of infection [15]. Elevated concentrations of IL-6 in amniotic fluid were reported in women with spontaneous preterm delivery [16]. In pregnant women with COVID-19, an immune activation was described at the maternal–fetal interface, even in the absence of detectable local viral invasion in COVID-19 during pregnancy [17]. SARS-CoV-2 enters the cell through angiotensin-converting enzyme (ACE2)/transmembrane protease, serine 2 (TMPRSS2), or by using the CD147/spike protein pathway [18,19]. ACE2, TMPRSS2, and CD147 are co-expressed in the syncytiotrophoblast, villous, and extravillous cytotrophoblast, while CD147 is expressed in amniotic epithelial cells. Therefore, the placenta and amniotic membranes may be infected. Placental and membrane inflammation can release inflammatory cytokines into fetal blood and amniotic fluid. Therefore, we have assumed that SARS-CoV-2 infection in pregnant women may induce inflammation within the amniotic cavity and/or an increase in proinflammatory cytokines in fetal circulation.

The aim of this study was to investigate IL-6 concentrations in maternal blood, umbilical cord blood, and amniotic fluid in pregnant women infected with SARS-CoV-2 at delivery.

## 2. Material and Methods

It was a single-center prospective observational case–control study of pregnant women diagnosed with SARS-CoV-2 infection, conducted between February and May 2022 (during the Omicron variant epidemic) at the Department of Obstetrics, Perinatology and Neonatology of Center of Postgraduate Medical Education. Women in singleton pregnancy, aged 18 years and older, with gestational age above 24 weeks, who delivered within 7 days following admission, were included in the study. A SARS-CoV-2 RT-qPCR test was performed in each patient upon admission to the hospital. SARS-CoV-2 RT-qPCR-positive women were included in the study group, while SARS-CoV-2 RT-qPCR-negative pregnant women matched for gestational age were included in the control group. All patients delivered within 7 days after the test performance via elective cesarean section. The following data were collected: age, body mass index (BMI), nicotine addiction, parity, twin pregnancy, preexisting hypertension, gestational hypertension, preeclampsia, and gestational diabetes mellitus (GDM). The BMI was calculated by dividing weight in kilograms by height in meters squared. Hypertension, preeclampsia, and diabetes mellitus were diagnosed according to the Polish Society of Obstetricians and Gynecologists’ guidelines [20]. A preterm delivery was defined as one occurring before 37 weeks of gestation. COVID-19 respiratory issues such as pneumonia, need for oxygen supplementation, need for continuous positive airway pressure (CPAP), and mechanical ventilation were analyzed.

Pregnant women with SARS-CoV-2 infection were monitored for symptoms of infection and dyspnoea. In cases of blood oxygen saturation below 95% on room air or a respiratory rate >30 breaths/min, oxygen supplementation was administered via a nasal canula or reservoir mask. All infected women were administered thromboprophylaxis. In case of mild respiratory syndromes, inhaled budesonide was administered, while in moderate and severe COVID-19, remdesivir within the first 5 days from the onset of infection and intravenous dexamethasone were given.

All women had white blood cell counts (WBC), serum C-reactive protein (CRP), procalcitonin (PCT), and IL-6 concentrations measured. During cesarean section, a 5 mL sample of amniotic fluid was collected with the use of a needle before opening the uterine cavity. After delayed cord clamping, a 5 mL sample of umbilical cord blood was collected. IL-6 concentrations in maternal and umbilical cord blood and amniotic fluid samples were measured with the VITROS 5600 immunodiagnostic system, using a two-stage immunometric technique with enhanced chemiluminescence reading. Viral RNA was isolated with the automated bioMérieux eMAG platform (nasopharyngeal swabs and umbilical cord blood) or a Macherey-Nagel RNA NucleoSpin Mini Kit (placenta tissue). An RT-qPCR for SARS-CoV-2 was performed with use of a bioMérieux SARS-CoV-2 R-Gene kit on a BioRad CFX96 cycler. Virological analyses were performed at the Chair and Department of Medical Microbiology, Medical University of Warsaw.

Neonatal birthweight and outcome, including the need for CPAP, mechanical ventilation, intraventricular hemorrhage (IVH), and neonatal intensive care unit (NICU) admission were analyzed. Composite neonatal morbidity was defined as the occurrence of at least one of the following: NICU admission, CPAP, mechanical ventilation, or IVH. Newborns small for gestational age (SGA) were diagnosed if their birthweight was below the 10th percentile for gestational age. SARS-CoV-2 RT-qPCR tests in nasopharyngeal swabs, umbilical cord blood, and placenta were performed in a subgroup of newborns delivered by mothers with SARS-CoV-2 infection.

The primary outcome was IL-6 concentrations in maternal, umbilical cord blood, and amniotic fluid in women with and without SARS-CoV-2 infection. Secondary outcomes comprised the analysis of the correlation between maternal, umbilical cord blood, and amniotic fluid IL-6 concentrations, and the relation between IL-6 concentrations and composite neonatal morbidity.

Assuming Pearson’s linear correlation r = 0.5 between the concentration of IL-6 in the pregnant woman’s blood and umbilical cord blood or amniotic fluid, with a power of 80% and type I error alpha = 0.05, the required size of the study and control groups was calculated to be 29. Variables were described as percentage or median (interquartile range). For statistical analysis, the Mann–Whitney test and the Fisher’s exact test were used. Box and whisker plots were created to visualize the results and are presented in Figure 1. *p*-values < 0.05 were considered significant. A Spearman’s analysis was performed to identify correlations between IL-6 concentrations. The data were analyzed using Statistica version 13.1, Tibco Software Inc., Palo Alto, CA, USA.

The study protocol was approved by the Bioethics Committee at the Center of Postgraduate Medical Education. The study was conducted in accordance with the World Medical Association Declaration of Helsinki. All enrolled women gave a written informed consent to participate in the study.

## 3. Results

A total of 48 women diagnosed with SARS-CoV-2 infection were included in the study. Their basic characteristics are presented in Table 1. They were matched with 48 women negative for SARS-CoV-2. Six women from the control group were excluded from the analysis due to the spontaneous onset of uterine contractions and cesarean section performed during the first stage of labor. Finally, there were 42 women in the control group. Their characteristics are presented in Table 1.

The study and control groups were similar in terms of age, BMI, and parity. Two women with COVID-19 required oxygen supplementation via nasal canula due to dyspnoea and blood oxygen saturation below 95% on room air. They were administered inhaled budesonide. None of the infected women required mechanical ventilation or CPAP. Less women in the study group were vaccinated for SARS-CoV-2. Although there were no differences in neonatal birthweights between the groups and all newborns were born in good general condition, six neonates in the study group (12.5%) suffered from respiratory disorders after delivery and were admitted to NICU. The comparison of outcomes between the study and the control group is presented in Table 1.

The values of WBC, CRP, PCT, and IL-6 are presented in Table 2. No significant differences were observed in maternal blood WBC, PCT, or IL-6 concentrations between SARS-CoV-2-positive and -negative women. CRP concentrations were higher in the study group, however, the difference was not significant. The concentrations of IL-6 in the amniotic fluid were similar in the study and control groups, while umbilical cord blood IL-6 concentrations were significantly higher in the SARS-CoV-2-positive women. There were no differences between maternal blood and umbilical cord blood IL-6 concentrations in the study group (*p* = 0.8), while IL-6 concentrations in the amniotic fluid were about 100-times higher (*p* < 0.01). Similar relations were found in the control group. The box and whisker plots of IL-6 concentrations in maternal blood, umbilical cord blood, and amniotic fluid are presented in Figure 1.

Further analysis of the correlations between IL-6 in maternal blood, umbilical cord blood, and amniotic fluid was performed. No correlation between maternal blood and amniotic fluid IL-6 concentrations, nor umbilical cord blood and amniotic fluid concentrations were found. An average correlation between maternal blood and umbilical cord blood IL-6 concentration was observed (r = 0.373, *p* = 0.01).

All newborns from the study group had nasopharyngeal swabs for SARS-CoV-2 RT-qPCR tests performed. In two cases, the test was positive—one directly after the delivery and one on the 7th day of life. IL-6 concentrations in the amniotic fluid and umbilical cord blood were as follows: >75,000 pg/mL and 25.9 pg/mL (SARS-CoV-2 infection diagnosed after delivery) and 1390 pg/mL and 4.19 pg/mL, respectively (SARS-CoV-2 infection diagnosed on the 7th day). SARS-CoV-2 RT-qPCR tests were performed in the umbilical cord blood and placenta of 17 newborns from the study group. None of the tests were positive. The newborn positive for SARS-CoV-2 directly after the delivery was born at 39 weeks of gestation weighing 4720 g in good general condition. The course of infection was asymptomatic. The other newborn was delivered at 39 weeks weighing 2890 g (8th percentile) in good general condition. The first PCR test was negative. Due to blood oxygen saturation of 85% during the first day of life he was administered oxygen therapy. The second PCR test, performed on the 7th day of life, was positive. The baby was discharged home in good general condition after 11 days of hospitalization.

The relation between IL-6 concentration and composite neonatal morbidity was analyzed. No significant differences were observed in maternal blood or amniotic fluid IL-6 concentrations, while IL-6 concentrations were significantly higher in the umbilical cord blood of newborns with adverse outcomes. The results are presented in Table 3.

## 4. Discussion

### 4.1. Maternal IL-6 Levels in Women Infected with SARS-CoV-2

This is the first study investigating IL-6 concentrations in blood, umbilical cord blood, and amniotic fluid in women with SARS-CoV-2 infection at delivery in comparison to healthy controls. Maternal concentrations of serum IL-6 were similar in SARS-CoV-2-positive and -negative women. This may be related to the fact that most of the included women were asymptomatic. Brancaccio et al. investigated pregnant women with symptomatic COVID-19 admitted to the COVID Unit of Mother and Child Department at University Hospital Federico II in 2021 and found a nearly three-fold increase in serum IL-6 concentrations in comparison to healthy controls [21]. Rubio et al. observed higher concentrations of serum IL-6 in pregnant women with COVID-19 than in blood of asymptomatic women infected with SARS-CoV-2 [22]. As IL-6 concentration is related to the severity of the disease, we observed moderately higher concentrations in infected women, however, the difference was not significant.

### 4.2. Amniotic Fluid and Umbilical Cord IL-6 Levels in Women Infected with SARS-CoV-2

We have investigated IL-6 concentrations in two compartments related to pregnancy—amniotic fluid and umbilical cord blood, which represent fetal circulation. We found significantly higher IL-6 concentrations in umbilical cord blood in SARS-CoV-2-positive women. Similar results were presented by Brancaccio et al. They found IL-6 in umbilical cord blood to be 3.6-times higher in infected women than in non-infected controls [21].

Maternal IL-6 levels were investigated in viral infections. Viremia may induce expression of IL-6 by fetal membranes and chorion, for example, in influenza infection [23]. High concentrations of IL-6 are present in amniotic fluid during the second and third trimester of pregnancy and increase in chorioamnionitis and preterm delivery [24,25]. In our study, around 100-times higher concentrations were detected in amniotic fluid both in SARS-CoV-2-positive and -negative women than in any blood samples taken in both groups. Significantly higher IL-6 concentrations in amniotic fluid compared to serum, both in maternal and umbilical cord blood, were also observed in a study by Santhanam et al. [25].

### 4.3. Relation between Maternal Blood, Umbilical Cord Blood, and Amniotic Fluid IL-6 Levels

To our knowledge no other study investigating IL-6 concentrations in active SARS-CoV-2 infection in maternal blood, umbilical cord blood, and amniotic fluid simultaneously and analyzing the relations between them has been published so far. Taglauer et al. reported on several cytokine concentrations in maternal and umbilical cord blood in 31 mother–newborn dyads, but their inclusion criterion was SARS-CoV-2 infection at any point during pregnancy [26]. The authors found significantly higher concentrations of serum IL-6 in infected women in both maternal and fetal blood in comparison to healthy controls. No correlations between maternal and fetal IL-6 concentrations were observed in that or other studies [7,25,26]. Taglauer et al. observed differing maternal–infant correlation profiles of IL-6, IL-8, and interferon-gamma-induced protein 10, which indicates that fetal cytokine elevations are not merely a passive transfer of maternal cytokines into fetal circulation [26]. Therefore, it can be assumed that maternal blood, amniotic fluid, and umbilical cord blood act like three separate compartments. The question is whether IL-6 concentrations in these compartments are interdependent or not. In animal models and in vitro placenta perfusions, a bidirectional transfer of IL-6 was observed [27,28]. Therefore, as IL-6 concentration in amniotic fluid is much higher than in blood, the passage from amniotic fluid into maternal and fetal circulation could be possible. Jongh et al. investigated the feto-maternal dependency of umbilical cord blood IL-6 concentrations in 81 healthy patients. Researchers collected maternal blood samples on admission to the hospital and just after the delivery. They found a significant positive correlation between neonatal cord blood and maternal serum IL-6 concentrations both on admission (r = 0.57, *p* < 0.001) and after the delivery (r = 0.79, *p* < 0.001). The correlation was not influenced by the mode of delivery [8]. These findings support the hypothesis that placental passage of IL-6 from the mother to the fetus is possible and are in accordance with our results. We observed a correlation between maternal blood and umbilical cord blood IL-6 concentrations. Another argument for placental passage of IL-6 was put forward by Singh et al. in a study investigating IL-6 concentrations in cord plasma and in neonatal mononuclear cells by reverse transcriptase-PCR before and after mitogenic stimulation. Elevated concentrations of IL-6 were found in cord plasma in chorioamnionitis, while mononuclear cells from neonates expressed no IL-6 mRNA in vivo. Cord blood mononuclear cells from healthy term babies were capable of synthesizing IL-6 in vitro in response to stimulation with bacterial lipopolysaccharide [29]. Therefore, it can be concluded that observed concentrations of IL-6 in cord blood are related to maternal levels of IL-6.

Proinflammatory cytokines can enter the uterine cavity and impair normal fetal development [30]. Elevated IL-6 concentration in pregnancy is related to the risk of miscarriage, preeclampsia, and preterm delivery [25,31,32]. According to our results, SARS-CoV-2 infection at delivery is related to higher concentrations of IL-6 in umbilical cord blood, even if the infection is asymptomatic. We found elevated IL-6 concentrations in cord blood to be related to composite neonatal morbidity as well. Proinflammatory cytokines and inflammation can impair the fetal nervous system. The injection of IL-6 into pregnant mice increased glutamatergic synapse density and disrupted hippocampal connectivity in offspring [33]. In humans, maternal immune system activation may influence fetal neurodevelopment and trigger autism spectrum disorder, attention deficit hyperactivity disorder, and cognitive dysfunction, and lead to anxiety, depression, and schizophrenia development later in life [34]. González-Mesa et al. investigated humoral and cellular immunity in maternal and umbilical cord blood in 79 pregnant women delivering after the diagnosis of SARS-CoV-2 infection during pregnancy. Although they found no SARS-CoV-2 virus RNA in any of the analyzed placental samples, detectable concentrations of IgG anti-SARS-CoV-2 antibodies, IL-1b, IL-6, and IFN-gamma were observed in umbilical cord blood. Significant correlations were discovered between IgG anti-SARS-CoV-2 antibodies and fetal CD3+ mononuclear cells, CD3+/CD4+, and CD3+/CD8+ T-cell subsets [35]. This proves that even without placental transmission of the virus, a significant fetal immune response is activated.

### 4.4. Strengths and Limitations

The strength of the presented research is its homogeneous study cohort of women infected with SARS-CoV-2 within a short time prior to delivery and a control group of non-infected women. All included participants had a SARS-CoV-2 RT-qPCR test performed. All women delivered via elective cesarean section. According to Treviño-Garza et al., IL-6 concentrations are affected by the mode of delivery, with an increase in concentration during vaginal birth [36]. Therefore, we have excluded a possible bias related to the mode of delivery. Sample collection of maternal blood, amniotic fluid, and umbilical cord blood in every mother–newborn dyad allowed us to analyze interdependencies of IL-6 concentrations between them. The study is, however, not free of limitations. One of them is the time of recruitment to the study, as between February and May 2022 the most common SARS-CoV-2 variant was Omicron, which was characterized by usually asymptomatic or mild symptomatic course. Only two women in our study group required oxygen therapy (4.2%). The oligosymptomatic course of infection probably influenced maternal IL-6 concentrations. Our cohorts were of moderate sample size, but unfortunately umbilical cord blood and placenta were tested for SARS-CoV-2 only in 17 cases.

## 5. Conclusions

Maternal SARS-CoV-2 infection in pregnant women at delivery increases umbilical cord blood IL-6 concentrations. The correlation between maternal and umbilical blood concentrations indicates a possibility of IL-6 passage through the placenta. This study supports a growing body of evidence that perinatal alterations resulting from maternal SARS-CoV-2 infection at delivery have a risk of impacting the health of infants even in the absence of fetal SARS-CoV-2 transmission or asymptomatic course of infection.

## Figures and Tables

**Figure 1 jcm-12-05672-f001:**
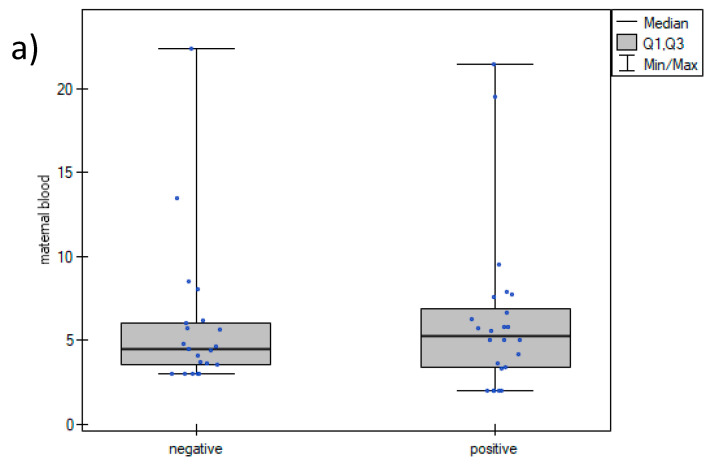
Box and whisker plots of IL-6 concentrations in maternal blood (**a**), umbilical cord blood (**b**), and amniotic fluid (**c**). Negative—group of SARS-CoV-2-negative women; positive—group of SARS-CoV-2-positive women, *—*p* = 0.02.

**Table 1 jcm-12-05672-t001:** Characteristics of the SARS-CoV-2-positive and -negative pregnant women and their newborns included in the study.

	SARS-CoV-2 Positive N = 48	SARS-CoV-2 Negative N = 42	
	Median (Interquartile Range)	Median (Interquartile Range)	*p*
Women			
Age (years)	32 (31–37)	33 (31–35.5)	1
BMI (kg/m^2^)	27.5 (25.5–33.4)	28 (26.4–30.3)	0.8
SARS-CoV-2 vaccination *	25 (52.1)	33 (78.6)	0.02
COVID-19 treated with oxygen supply *	2 (4.2)	-	
COVID-19 pneumonia	0	-	
CPAP	0	-	
Mechanical ventilation	0	-	
Primiparous *	18 (37.5)	18 (42.9)	0.6
Chronic hypertension *	2 (4.2)	1 (2.4)	0.8
Gestational hypertension/preeclampsia *	6 (12.5)	1 (2.4)	0.1
SGA *	5 (10.4)	4 (9.5)	0.9
Gestational diabetes mellitus *	3 (6.3)	4 (9.5)	0.8
Nicotinism in pregnancy *	3 (6.3)	0	0.1
Gestational age at delivery (weeks)	38 (37–39)	39 (38–39.25)	0.3
Preterm delivery *	7 (14.5)	2 (4.8)	0.2
Newborns			
Birthweight (g)	3375 (3110–3910)	3510 (3092–3742)	0.7
Apgar < 8 *	0	0	1
NICU admission *	6 (12.5)	0	0.046
CPAP *	6 (12.5)	0	0.046
Mechanical ventilation *	1 (2.1)	0	0.9
IVH *	0	0	1

*—number (percent)**,** BMI—body mass index; SGA—small for gestational age; NICU—neonatal intensive care unit; CPAP—continuous positive airway pressure; IVH—intraventricular hemorrhage.

**Table 2 jcm-12-05672-t002:** Laboratory parameters of infection in women with and without SARS-CoV-2.

		SARS-CoV-2 Positive N = 48	SARS-CoV-2 Negative N = 42	
		Median (Interquartile Range)	Median (Interquartile Range)	*p*
Maternal blood	WBC (10^3^/μL)	9.4 (6.7–11.3)	9.1 (8–10.6)	0.9
	CRP (mg/L)	6.9 (2.6–15.6)	2.8 (1.5–3.9)	0.06
	PCT (ng/mL)	0.07 (0.04–0.1)	0.04 (0.01–0.07)	1
	IL-6 (pg/mL)	5.3 (3.4–7.1)	4.4 (3–5.9)	0.6
Amniotic fluid	IL-6 (pg/mL)	511 (259–880.5)	514 (310–770.5)	0.9
Umbilical cord blood	IL-6 (pg/mL)	6.2 (4.3–8.3)	4.2 (2.6–4.9)	0.02

WBC—white blood cell count; CRP—C-reactive protein; PCT—procalcitonin, IL-6—interleukin 6.

**Table 3 jcm-12-05672-t003:** The relation between IL-6 concentration and composite neonatal morbidity.

	Composite Neonatal Morbidity	No Composite Neonatal Morbidity	
IL-6 (pg/mL)	Median (Interquartile Range)	Median (Interquartile Range)	*p*
Maternal blood	5.1 (4.14–9.64)	4.5 (3.38–7.34)	0.4
Amniotic fluid	294 (168–733)	524 (323–860)	0.4
Umbilical cord blood	7.6 (5.02–9.54)	4.6 (3–6.22)	0.03

IL-6—interleukin 6.

## Data Availability

Data will be available on request.

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
