# Peer review of "Maternal SARS-CoV-2 Infection at Delivery Increases IL-6 Concentration in Umbilical Cord Blood"

_jcm, 2023, doi:10.3390/jcm12175672_

Round 1

Reviewer 1 Report

The article "Maternal SARS-CoV-2 infection at delivery increases IL-6 concentration in umbilical cord blood" presents new knowledge about the possible transmission of the pro-inflammatory cytokine IL-6 to the blood of newborns and its presence in umbilical cord blood.  

The authors conclude that maternal SARS-CoV-2 infection in pregnant women at delivery increases umbilical cord blood IL-6 concentrations. The correlation between maternal and umbilical blood concentrations indicates a possibility of IL-6 passage through the placenta. This study supports a growing body of evidence that perinatal alterations resulting from maternal SARS-CoV-2 infection at delivery risk impact the health of infants even in the absence of fetal SARS-CoV-2 transmission or asymptomatic course of infection.

The methods are well designed and the results and discussion are well described!

Despite that, there are some remarks in the introduction that need to be corrected and messing information in Material and Methods to add:

1-    Line 52-57: Authors must be careful and reconsider this paragraph because there is no evidence of vertical transmission until now for SARS-COV-2.

2-    Line 101-112: Authors explain perfectly the methods they have used to measure IL6 concentration in maternal blood and amniotic fluid samples but there is no information for the method to measure IL6 concentration in umbilical cord blood,

If a different technique has been used, authors have to mention it.

The article brings new information on the topic of SARS-COV-2 infection in pregnant women and the study could be very helpful in understanding that even if the infection is asymptomatic in pregnant women, babies born from infected mothers have to be followed to their growing and healthy, especially for their neurodevelopment that we know that inflammation could be very harmful to the brain development at early stage.

Author Response

               We thank the Reviewer for the positive review and constructive comments. We have revised the manuscript according to the suggestions and we hope that the changes will convince the Reviewers and the Editor that the paper is worthy of publication in Journal of Clinical Medicine.              

Here are our responses to Reviewer’s comments:

1.  We have revised the paragraph into less categorical tone changing “proves that vertical transmission from the mother to the fetus is possible” into “the potential of congenital, intrapartum, and postnatal maternal-fetal-neonatal SARS-CoV-2 infections”.    

2. The method of IL-6 concentration measurement in both maternal and umbilical blood was identical. This information was added to the manuscript.

Reviewer 2 Report

This may be an important study in case authors are able to show data to support association of increased IL-6 levels in cord blood, maternal blood and adverse outcome including fatal mortality. Was any adverse effect like IUGR noted?

There are too many tables which can be reduced if possible without compromising on clarity of data.

In discussion I will like to see; what studies done of IL-6 in pregnant women with infection other than Covid-19 are reporting. If there are no other studies please mention. 

It will also be interesting to see the applied value of this finding.

Discussion needs revision because it is unorganised and difficult to comprehend.

Acceptable

Author Response

We thank the Reviewer for the positive review and constructive comments. We have revised the manuscript according to the suggestions and we hope that the changes will convince the Reviewers and the Editor that the paper is worthy of publication in Journal of Clinical Medicine.              

Here are our responses to Reviewer’s comments:

  1.     “This may be an important study in case authors are able to show data to support association, maternal blood and adverse outcome including fatal mortality. Was any adverse effect like IUGR noted?”We did not observe maternal blood IL-6 level to be significantly elevated in women infected with SARS-CoV-2. This may be due to the fact that most of women in our study had asymptomatic or mild course of disease. No differences in neonatal birthweight nor newborns small for gestational age rates were observed between the groups of infected and non-infected women. There were no cases of fetal demise, maternal or neonatal death. We assume that  the relation between IL-6 level in umbilical cord blood and composite neonatal morbidity is due to more frequent respiratory disorders in newborns from mothers infested with SARS-CoV-2.
  2.      There are too many tables which can be reduced if possible without compromising on clarity of data.We have revised the manuscript according to this suggestion and removed one table.
  3.      In discussion I will like to see; what studies done of IL-6 in pregnant women with infection other than Covid-19 are reporting. If there are no other studies please mention. IL-6 was investigated in other infections like influenza in pregnancy women. It was added to the manuscript.
  4.      It will also be interesting to see the applied value of this finding.We agree absolutely and intend to perform another prospective study of IL-6 concentrations in maternal and umbilical cord blood in other infections and pregnancy complications like preeclampsia.
  5.      Discussion needs revision because it is unorganised and difficult to comprehend.Discussion has been revised and divided into separate sections to increase the clarity of the text.